# Fly-QMA: Automated analysis of mosaic imaginal discs in *Drosophila*

**Sebastian M. Bernasek**[1,2], **Nicolás Peláez**[3¤], **Richard W. Carthew**[2,3,4],
**Neda Bagheri**[1,2,5,6,7]*, **Luís A. N. Amaral**[1,2,7,8]*

**1** Department of Chemical and Biological Engineering, Northwestern University, Evanston, Illinois, United States of America, **2** NSF-Simons Center for Quantitative Biology, Northwestern University, Evanston, Illinois, United States of America, **3** Department of Molecular Biosciences, Northwestern University, Evanston, Illinois, United States of America, **4** Department of Biochemistry and Molecular Genetics, Northwestern University, Evanston, Illinois, United States of America, **5** Department of Biology, University of Washington, Seattle, Washington, United States of America, **6** Department of Chemical Engineering, University of Washington, Seattle, Washington, United States of America, **7** Northwestern Institute on Complex Systems, Northwestern University, Evanston, Illinois, United States of America, **8** Department of Physics and Astronomy, Northwestern University, Evanston, Illinois, United States of America

¤ Current address: Division of Biology and Biological Engineering, California Institute of Technology, Pasadena, California, United States of America
* nbagheri@uw.edu (NB); amaral@northwestern.edu (LANA)

**Data Availability Statement:** The data underlying the results presented in the study are available in a public data repository hosted by Northwestern University. DOI: https://doi.org/10.21985/N2F207.

## Abstract

Mosaic analysis provides a means to probe developmental processes in situ by generating loss-of-function mutants within otherwise wildtype tissues. Combining these techniques with quantitative microscopy enables researchers to rigorously compare RNA or protein expression across the resultant clones. However, visual inspection of mosaic tissues remains common in the literature because quantification demands considerable labor and computational expertise. Practitioners must segment cell membranes or cell nuclei from a tissue and annotate the clones before their data are suitable for analysis. Here, we introduce Fly-QMA, a computational framework that automates each of these tasks for confocal microscopy images of *Drosophila* imaginal discs. The framework includes an unsupervised annotation algorithm that incorporates spatial context to inform the genetic identity of each cell. We use a combination of real and synthetic validation data to survey the performance of the annotation algorithm across a broad range of conditions. By contributing our framework to the open-source software ecosystem, we aim to contribute to the current move toward automated quantitative analysis among developmental biologists.

## Author summary

Biologists use mosaic tissues to compare the behavior of genetically distinct cells within an otherwise equivalent context. The ensuing analysis is often limited to qualitative insight. However, it is becoming clear that quantitative models are needed to unravel the complexities of many biological systems. In this manuscript we introduce a computational framework that automates the quantification of mosaic analysis for Drosophila imaginal discs, a common setting for studies of developmental processes. The software extracts quantitative

**Funding:** SMB and LANA were supported by the John and Leslie McQuown Gift. RWC was supported by NIH R35GM118144 (https://www.nih.gov). LANA, NB, and RWC were supported by NSF 1764421 (https://www.nsf.gov). LANA, NB, and RWC were supported by Simons Foundation 597491 (https://www.simonsfoundation.org). NP was supported by the HHMI Hanna H. Gray Fellowship (https://www.hhmi.org/programs/hanna-h-gray-fellows-program). In all cases, the funders had no role in study design, data collection and analysis, decision to publish, or preparation of the manuscript.

**Competing interests:** The authors have declared that no competing interests exist.

measurements from confocal images of mosaic tissues, rectifies any cross-talk between fluorescent reporters, and identifies clonally-related subpopulations of cells. Together, these functions allow users to rigorously ascribe changes in gene expression to the presence or absence of particular genes. We validate the performance of our framework using both real and synthetic data. We invite interested readers to apply these methods using our freely available software.

## Introduction

Quantification will be essential as biologists study increasingly complex facets of organismal development [1]. Unfortunately, qualitative analysis remains common because it is often difficult to measure cellular processes in their native context. Modern fluorescent probes and microscopy techniques make such measurements possible [2–4], but the ensuing image analysis demands specialized skills that fall beyond the expertise of most experimentalists. Automated analysis strategies have addressed similar challenges in cytometry [5–7], genomics and transcriptomics [8–11], and other subdisciplines of biology [12, 13]. Image analysis has proven particularly amenable to automation, with several computer vision tools having gained traction among biologists [14–17]. These platforms are popular because they increase productivity, improve the consistency and sensitivity of measurements, and obviate the need for specialized computational proficiency [18–20]. Designing similar tools to help biologists probe and measure developmental processes in vivo will further transform studies of embryogenesis and development into quantitative endeavors.

Developmental biologists study how the expression and function of individual genes coordinate the emergence of adult phenotypes. They often ask how cells respond when a specific gene, RNA, or protein is perturbed during a particular stage of development. Cell response may be characterized by changes in morphology, or by changes in the expression of other genes (Fig 1A). Experimental efforts to answer this question were historically stifled by the difficulty of isolating perturbations to a single developmental context, as the most interesting perturbation targets often confer pleiotropic function across several stages of development and can trigger early embryonic lethality [21–23].

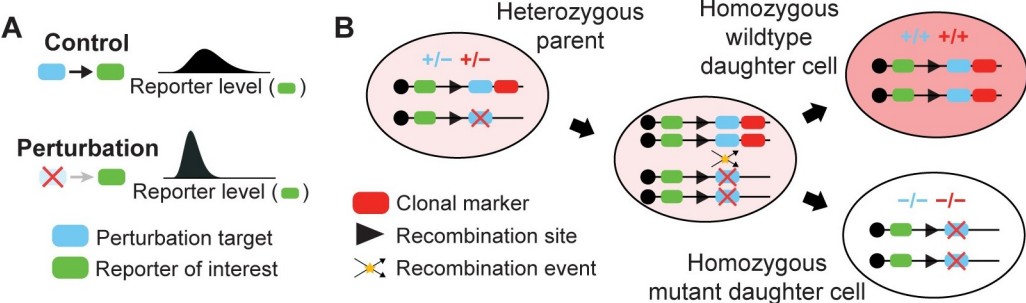

**Fig 1. Perturbing gene expression via mitotic recombination.** Experimental framework using mitotic clones to test whether or not regulatory interactions occur between a perturbation target and reporter of interest. Blue and green markers represent the respective genes encoding the perturbation target and the reporter. (A) A perturbation-induced decrease in reporter levels would confirm that regulation occurs. (B) Mitotic recombination generates clonal subpopulations carrying zero, one, or two copies of the gene encoding a perturbation target. Black lines depict a genetic locus. Only genes downstream of the recombination site are subject to recombination. Red markers represent a gene encoding a clonal marker used to identify the resultant clones. Red shading of large oval reflects relative clonal marker fluorescence level.

Mosaic analysis addressed this challenge in *Drosophila* by limiting perturbations to a subset of cells within the imaginal discs of the larva [24, 25]. The technique yields a heterogeneous tissue comprised of genetically distinct patches of cells that are clonally related. Aside from rare de novo mutations, cells within each clone are genetically identical. Clone formation may be restricted to specific developing organs by using disc-specific gene promoters to drive transchromosomal recombination events in the corresponding imaginal discs [26, 27]. The timing of these events determines the number and size of the resultant clones [28]. Perturbations are applied by engineering the dosage of a target gene to differ across clones (Fig 1B), resulting in clones whose cells are either homozygous mutant (−/−), heterozygous wildtype (+/−), or homozygous wildtype (+/+) for the particular gene. Labeling these clones with the presence or absence of fluorescent markers enables direct comparison of cells subject to control or perturbation conditions, while maintaining otherwise equivalent developmental and physiological histories between the two cell populations (Fig 2A). Additional reporters may be used to monitor differences in RNA or protein expression, morphology, or cell fate choice across clones (Fig 2B). Variants of this strategy led to seminal discoveries in both neural patterning [29–31] and morphogenesis [32, 33], and remain popular today [34–36].

Quantitative microscopy techniques are well suited to measuring differences in cell behavior across clones. One reporter (a clonal marker) labels the clones, while others quantitatively report properties of their constituent cells, such as the expression level of a gene product of interest (Fig 2C). The former then defines the stratification under which the latter are compared. We call this strategy Quantitative Mosaic Analysis (QMA) because it replaces subjective visual comparison with a rigorous statistical alternative. Although a few recent studies have deployed this approach [37–40], qualitative visual comparison remains pervasive in the literature.

We suspect the adoption of QMA has been hindered by demand for specialized computational skills or, in their stead, extensive manual labor. Researchers must first draw or detect boundaries around individual nuclei in a procedure known as segmentation (Fig 2D). Averaging the pixel intensities within each boundary then yields a fluorescence intensity measurement for each reporter in each identified nucleus (Fig 2E). The measurements should then be corrected to account for any fluorescence bleedthrough between reporter channels (Fig 2F). Correction often requires single-reporter calibration experiments to quantify any potential crosstalk between different fluorophores, followed by complex calculations to remedy the data [41, 42]. Researchers must then label, or annotate, each identified nucleus as mutant, heterozygous, or homozygous for the clonal marker. Annotation is typically achieved through visual inspection (Fig 2G). Cells carrying zero, one, or two copies of the clonal marker should exhibit low, medium, or high average levels of fluorescence, respectively. However, both measurement and biological noise introduce the possibility that some cells' measured fluorescence levels may not reliably reflect their genetic identity. Annotation must therefore also consider the spatial context surrounding each nucleus. For instance, a nucleus whose neighbors express high levels of the clonal marker is likely to be homozygous for the clonal marker, even if its individual fluorescence level is comparable to that of heterozygous cells (Fig 2G, white arrows). Spatial context is particularly informative in developing tissues where cell migration is minimal, such as the fly imaginal discs. With many biological replicates containing thousands of cells each, annotation can quickly become insurmountably tedious. The corrected and labeled measurements are then curated for statistical comparison by excluding those on the border of each clone, and limiting their scope to particular regions of the image field (Fig 2H). Combined, all of these tasks ultimately burden researchers and raise the barrier for adoption of QMA.

Automation promises to alleviate this bottleneck, yet the literature bears surprisingly few computational resources designed to support QMA. The ClonalTools plugin for ImageJ

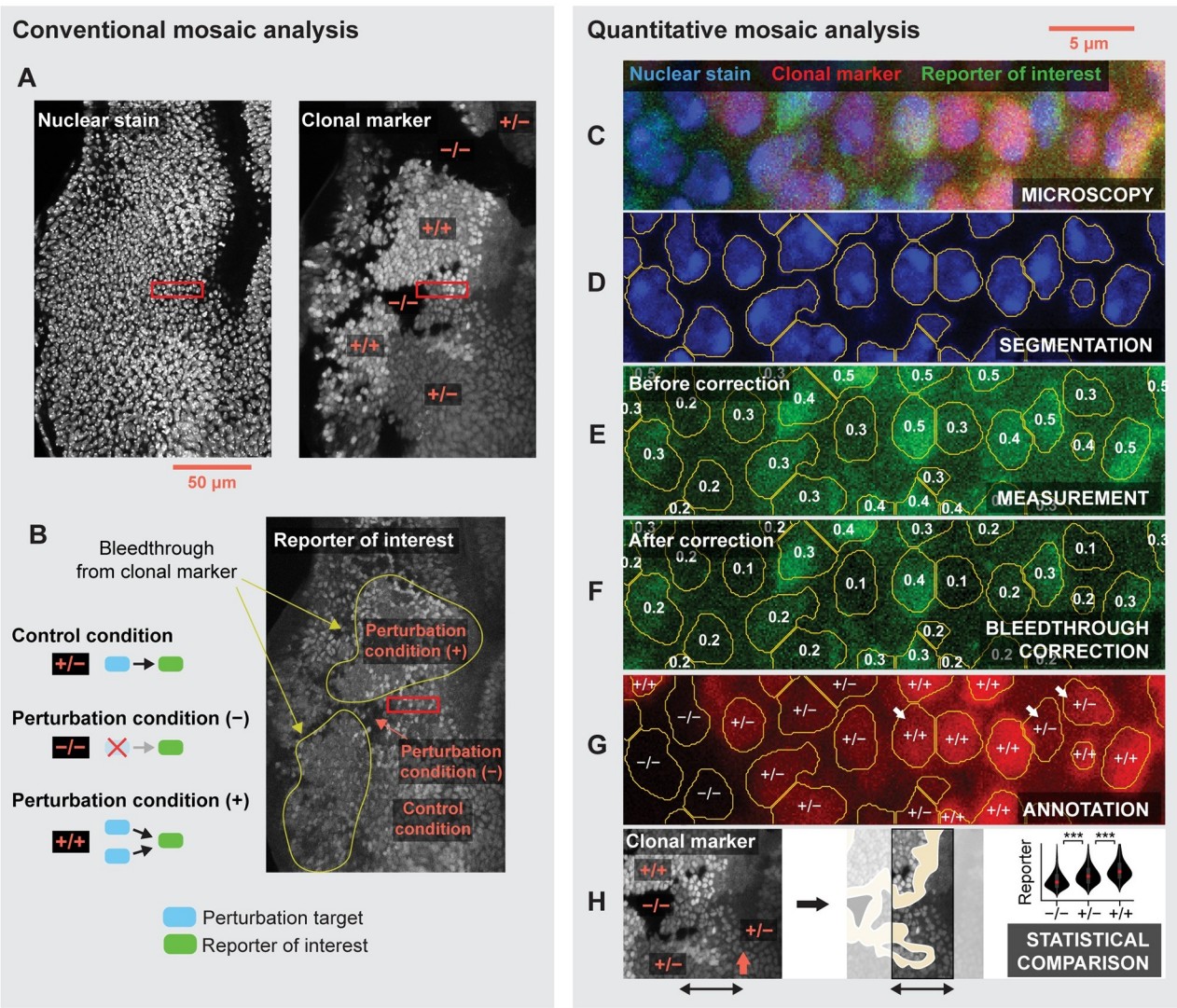

**Fig 2. Conventional versus quantitative mosaic analysis.** (A,B) Conventional analysis of a mosaic eye imaginal disc. (A) Clones are identified by visual comparison of clonal marker fluorescence among nuclei. (B) Regions labeled homozygous mutant (−/−) or homozygous wildtype (+/+) for the clonal marker are compared with those labeled heterozygous wildtype (+/−) to assess whether reporter expression differs across clones. Fluorescence bleed-through is arbitrarily diagnosed. (C-H) Quantitative mosaic analysis. Panels depict a magnified view of the region enclosed by red rectangles in panels A and B. (C) Raw confocal image of the nuclear stain, clonal marker, and reporter of interest. (D) Segmentation identifies distinct nuclei. (E) Reporter expression is quantified by averaging the pixel intensities within each segment. Numbers reflect measured values. (F) Measurements may be corrected to mitigate fluorescence bleedthrough. (G) Individual nuclei are labeled homozygous mutant, heterozygous, or homozygous wildtype for the clonal marker. White arrows mark nuclei with ambiguous fluorescence levels. (H) Reporter levels are compared across clones to determine whether the perturbation affects reporter expression. Yellow region marks excluded clone borders. Comparison may exclude clone borders (yellow regions) and focus on a particular region of the image field (black arrows). In the eye imaginal disc, comparison is often limited to a narrow window near the MF (orange arrow).

deploys an image-based approach to measure macroscopic features of clone morphology, but is limited to binary classification of mutant versus non-mutant tissue and offers no functionality for comparing reporter expression across clones [43]. Alternatively, the MosaicSuite plugin for ImageJ deploys an array of image processing, segmentation, and analysis capabilities to automatically detect spatial interactions between objects found in separate fluorescence channels [44, 45]. While useful in many other settings, neither of these tools support automated labeling of individual cells or explicit comparison of clones with single-cell resolution. Most

modern studies employing a quantitative mosaic analysis instead report using some form of ad hoc semi-automated pipeline built upon ImageJ [37, 39, 40]. We are therefore unaware of any platforms that offer comprehensive support for an automated QMA workflow.

Here, we introduce Fly-QMA, a computational framework for automated QMA of *Drosophila* imaginal discs. Fly-QMA supports segmentation, bleedthrough correction, and annotation of confocal microscopy data (Fig 2D–2H). We demonstrate each of these functions by applying them to real confocal images of clones in the eye imaginal disc, and find that our automated approach yields results consistent with manual analysis by a human expert. We then generate and use synthetic data to survey the performance of our framework across a broad range of biologically plausible conditions. Fly-QMA is freely available online (see Data and software availability), along with an interactive coding tutorial designed to acquaint users with the core software features by applying them to example data.

## Results

### Quantification of nuclear fluorescence levels

We implemented a segmentation strategy based upon a standard watershed approach [52]. Briefly, we construct a foreground mask by Otsu thresholding the nuclear stain or nuclear label image following a series of smoothing and contrast-limited adaptive histogram equalization operations [52, 53]. We then apply a Euclidean distance transform to the foreground mask, identify the local maxima, and use them as seeds for watershed segmentation. When applied to the microscopy data, few visible spots in the nuclear stain were neglected, and the vast majority of segments outlined individual nuclei (S1C Fig).

This approach is flexible and should perform adequately in many scenarios. However, we acknowledge that no individual strategy can address all microscopy data because segmentation is strongly context dependent. All subsequent stages of analysis were therefore designed to be compatible with any data that conform to our standardized file structure. This modular arrangement grants users the freedom to use one of the many other available segmentation platforms [54], including FlyEye Silhouette [55], before applying the remaining functionalities of our framework. Regardless of how nuclear contours are identified, averaging the pixel intensities within them yields fluorescence intensity measurements for each reporter in each identified nucleus. We next sought to ensure that these measurements were suitable for comparison across clones.

### Bleedthrough correction

Despite efforts to select non-overlapping reporter bandwidths and excite them sequentially, it is not uncommon for reporters excited at one wavelength to emit some fluorescence in the spectrum collected for another channel (Fig 2B, yellow lines) [41, 56]. The end result is a positive correlation, or crosstalk, between the measured fluorescence intensities of two or more reporters. Exogenous correlations between the measured fluorescence intensities of the clonal marker and the reporter of interest are problematic given that the purpose of the experiment is to detect changes in reporter levels with respect to the clonal marker.

In our microscopy data, individual clones were distinguished by their low, medium, or high expression levels of an RFP-tagged clonal marker (Fig 3A). These images should not have shown any detectable difference in GFP levels across clones because all cells carried an equivalent dosage of the control reporter (S1A Fig). However, the images visibly suffered from bleedthrough between the RFP and GFP channels (Fig 3A and 3B). Bleedthrough was similarly evident when we compared measured GFP levels across labeled clones. Nuclei labeled mutant, heterozygous, or homozygous for the clonal marker had low, medium, and high expression

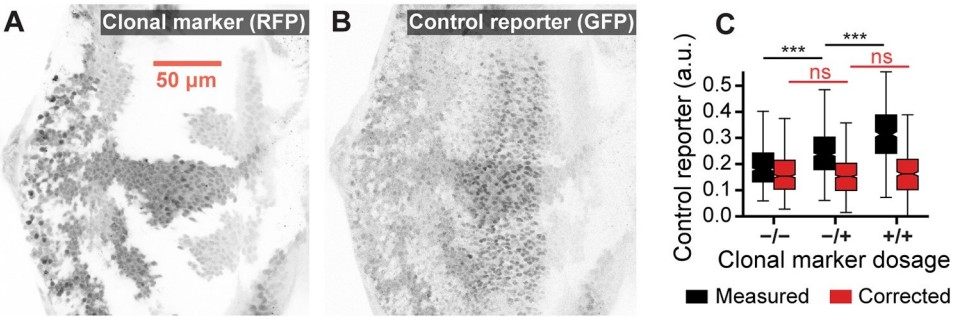

**Fig 3. Automated correction of fluorescence bleedthrough in the larval eye.** (A) Low, medium, and high expression levels of the RFP-tagged clonal marker. (B) GFP-tagged control reporter expression. RFP fluorescence bleedthrough is visually apparent upon comparison with A. (C) Comparison of control reporter expression between clones. Includes data aggregated across nine images taken from six separate eye discs. Data were limited to cells within the region of elevated GFP expression that were of approximately comparable developmental age (see S2E–S2G Fig). Measurements are stratified by their assigned labels. Before correction, expression differs between clones (black boxes, $p < 10^{-5}$). No difference is detected after correction (red boxes, $p > 0.05$).

levels of the control reporter, respectively (Fig 3C, black boxes). The data were therefore ripe for systematic correction.

Spectral bleedthrough correction is common practice in other forms of cross-correlation and co-localization microscopy [41, 56]. These methods typically entail characterizing the extent of crosstalk between fluorophores globally [57, 58], on a pixel-by-pixel basis [42], or by experimental calibration [41], then detrending all images or measurements prior to subsequent analysis. Our framework adopts the global approach, using the background pixels in each image to infer the extent of fluorescence bleedthrough across spectral channels.

Specifically, we assume the fluorescence intensity $F_{ij}$ for channel $i$ at pixel $j$ is a superposition of a background intensity $B_{ij}$ and some function of the expression level $E_{ij}$ that we seek to compare across cells [59]:

$$F_{ij} = B_{ij} + f(E_{ij}) \tag{1}$$

We further assume that the background intensity of a channel includes linear contributions from the fluorescence intensity of each of the other channels:

$$B_{ij} = \sum_{k \neq i} \alpha_k F_{kj} + \beta \tag{2}$$

where $k$ is indexed over $K$ anticipated sources of bleedthrough. Given estimates for each $\{\alpha_1, \alpha_2, \ldots \alpha_K\}$ and $\beta$ we can then estimate the background intensity of each measurement:

$$\langle B_{ij} \rangle = \sum_{k \neq i} \alpha_k \langle F_{kj} \rangle + \beta \tag{3}$$

where the braces denote the average across all pixels within a single nucleus. The corrected signal value is obtained by subtracting the background intensity from the measured fluorescence level:

$$\langle f(E_{ij}) \rangle = \langle F_{ij} \rangle - \langle B_{ij} \rangle \tag{4}$$

Repeating this procedure for each nucleus facilitates comparison of relative expression levels across nuclei in the absence of bleedthrough effects. Bleedthrough correction performance

is therefore strongly dependent upon accurate estimation of the bleedthrough contribution strengths, $\{\alpha_1, \alpha_2, \ldots \alpha_K\}$.

We estimate these parameters by characterizing their impact on background pixels (see Methods). When applied to the microscopy data, bleedthrough correction successfully eliminated any detectable difference in GFP expression across clones (Fig 3C, red boxes, $p > 0.05$ two-sided Mann-Whitney $U$ test).

## Automated annotation of clones

Our annotation strategy seeks to label each identified cell as homozygous mutant, heterozygous wildtype, or homozygous wildtype for the clonal marker. Variation within each clone precludes accurate classification of a cell's genotype solely on the basis of its individual expression level. However, in tissues where cell migration is minimal, clonal lineages are unlikely to exist in isolation because recombination events are typically timed to generate large clones. Our strategy therefore integrates both clonal marker expression and spatial context to identify clusters of cells with locally homogeneous expression behavior, then maps each cluster to one of the possible labels. This unsupervised approach lends itself to automated annotation because the clusters are inferred directly from the data without any guidance from the user.

We first train a statistical model to estimate the probability that a given measurement came from a cell carrying zero, one, or two copies of the clonal marker (S3A Fig). This entails fitting a weighted mixture of three or more bivariate lognormal distributions (components) to a two dimensional set of observations (S3B and S3C Fig). The first dimension corresponds to the clonal marker fluorescence level measured within each cell. The second dimension describes the local average expression level within the region surrounding each cell. We evaluate the latter by estimating a neighborhood radius from the decay of the radial correlation of the expression levels, then averaging the expression levels of all cells within that radius (S3D Fig). The second dimension therefore measures the spatial context in which a cell resides. We balance model fidelity against overfitting by using the Bayesian information criterion to determine the optimal number of model components (S3E Fig). We then cluster the components into three groups on the basis of their mean values (S3F Fig), effectively mapping each component to one of the three possible gene dosages. The model may be trained using observations derived from a single image, or with a collection of observations derived from multiple images. Once trained, the model is able to predict the conditional probability that an individual observation belongs to one of the model's components, given its measured expression level.

We then use the learned conditional probabilities to detect entire clones, thus assigning a label to each cell. Rather than using the trained model to classify each observation, we compile a new set of observations by limiting each estimate of spatial context to spatially collocated communities with similar expression behavior (S4A Fig). We identify these communities by applying a community detection algorithm to an undirected graph connecting adjacent cells (S4B Fig). Edges in this graph are weighted by the similarity of clonal marker expression between neighbors, resulting in communities with similar expression levels (S4E Fig, Steps I and II). The graph-based approach increases spatial resolution by limiting the information shared by dissimilar neighbors. Applying the mixture model yields an initial estimate of the probability that an observation belongs to one of the model's components (S4E Fig, Step III). We further refine these estimates by allowing the probabilities estimated for each cell to diffuse throughout the graph (S4E Fig, Step IV). The rate of diffusion between neighbors is determined by the weight of the edge that connects them, with more similar neighbors exerting stronger influence on each other. We then use the diffused probabilities to identify the most probable source component and label each observation (S4E Fig, Step V). These probabilities

also provide a measure of confidence in the assigned labels. We replace any low-confidence labels with alternate labels assigned using a marginal classifier that neglects spatial context (S4F and S4G Fig), resulting in a fully labeled image (S4H Fig).

The algorithm leverages the collective wisdom of neighboring measurements to override spatially isolated fluctuations in clonal marker expression, and thereby enforces consistent annotation within contiguous regions of the image field. The size of these regions depends upon the granularity of estimates for the spatial context surrounding each cell. We used an unsupervised approach to choose an appropriate spatial resolution in a principled manner. In short, the resolution is matched to the approximate length scale over which expression levels remain correlated among cells. Both the training and application stages of our annotation algorithm use this automated approach (S3D and S4D Figs), thus averting any need for user input.

## Manual assessment of annotation performance

We sought to validate the performance of the annotation algorithm by assessing its ability to accurately reproduce human-assigned labels. We manually labeled nuclei in each eye imaginal disc as homozygous mutant, heterozygous wildtype, or homozygous wildtype for the clonal marker, then automatically labeled the same cells (Fig 4A). The two sets of labels showed strong overall agreement (Fig 4B and S5A Fig). Excluding cells on the border of each clone revealed greater than 97% agreement in seven of the nine annotated images (see Table 1). Upon secondary inspection of the sole instance of substantial disagreement (S5B Fig), we are unable to confidently discern which set of labels are more accurate. While manual labeling required more than one hour of labor per image, the annotation algorithm achieved comparable accuracy in a matter of seconds. This performance advantage would continue to grow if the analysis were extended to multiple image layers, tissue samples, and experimental conditions.

While it is common practice to use human-labeled data as the gold standard, manually assigned labels do not represent a reliable and reproducible ground truth. Furthermore, we contend that validation with manually-labeled data entrains implicit human biases in the

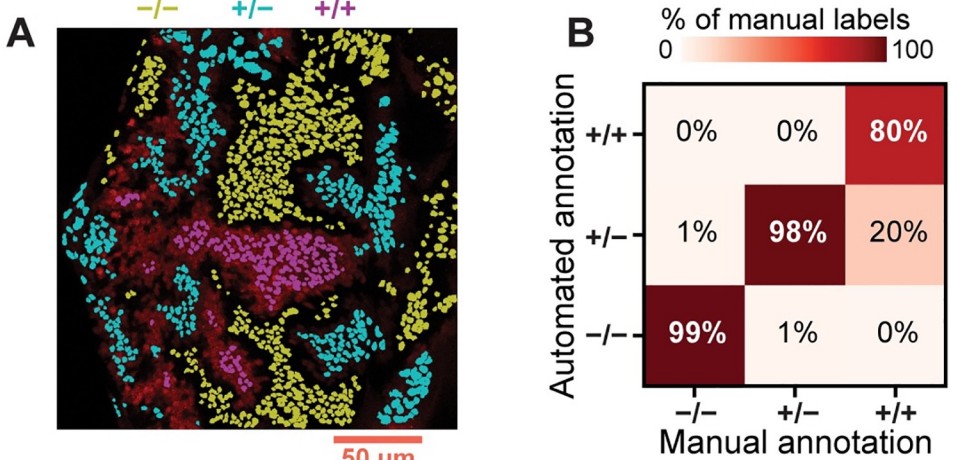

**Fig 4. Automated unsupervised annotation of clones in the larval eye.** (A) Labels assigned by automated annotation. Yellow, cyan, and magenta denote the label assigned to each contour. Labels are overlayed on the RFP channel of the image shown in S1B Fig. Cells on the periphery of each clone are excluded. (B) Comparison of automated annotation with manually-assigned labels. Confusion matrix includes data aggregated across nine images taken from six separate eye discs. Cells on the periphery of each clone are excluded. Columns sum to one.

**Table 1. Automated vs. manual annotation.**

| Disc | Layer | Agreement* |
|------|-------|-----------|
| 1 | 1 | 93.1% (97.3%) |
| 1 | 2 | 95.3% (97.3%) |
| 2 | 1 | 91.3% (99.1%) |
| 2 | 2 | 95.2% (96.4%) |
| 3 | 1 | 67.2% (75.6%) |
| 4 | 1 | 82.5% (89.2%) |
| 5 | 1 | 96.2% (100%) |
| 6 | 1 | 99.1% (99.3%) |
| 6 | 2 | 95.2% (97.5%) |

* Values in parentheses denote agreementwhen clone borders are excluded.

selection of performant algorithms. These biases are particularly pronounced in biological image data where intrinsic variation, measurement noise, and transient processes can make cell-type annotation a highly subjective, and thus irreproducible, task.

## Synthetic benchmarking of annotation performance

Synthetic benchmarking provides a powerful alternative to validation against manually labeled data. The idea is simple; measure how accurately an algorithm is able to label synthetic data for which the labels are known. The synthetic data generation procedure may be modeled after the process underlying formation of the real data, providing a means to assess the performance of an algorithm across the range of conditions that it is likely to encounter. The strategy therefore provides a means to survey the breadth of biologically plausible conditions under which the algorithm provides adequate performance. Synthetic benchmarking also facilitates unbiased comparison of competing algorithms, resulting in a reliable standard that may be called upon at any time.

We used synthetic microscopy data to benchmark the performance of our annotation strategy. Each synthetic dataset depicts a simulated culture of cells distributed roughly uniformly in space (S6A Fig). Cells in this culture contain zero, one, or two copies of a gene encoding an RFP-tagged clonal marker (S6B Fig). Our simulation procedure ensures that cells tend to remain proximal to their clonal siblings (S6C Fig), thus forming synthetic clones with tunable size and spatial heterogeneity (S6D and S6E Fig). We generated synthetic measurements by randomly sampling fluorescence levels in a dosage-depend manner (S7A–S7C Fig). We varied the similarity of fluorescence levels across clones using an ambiguity parameter, $\sigma_\alpha$, that modulates the spread of the distributions used to generate fluorescence levels (S7D–S7F Fig).

Using this schema as a template, we generated a large synthetic dataset, annotated each set of measurements, and compared the assigned labels with their true values. We used the mean absolute error as a comparison metric because it provides a stable measure of accuracy for multiclass classification problems in which the labels are intrinsically ordered [60]. In other words, it penalizes egregious misclassifications more severely than mild ones.

Annotation performance is very strong for all cases in which $\sigma_\alpha \leq 0.3$ (Fig 5). Unsurprisingly, performance suffers as the difficulty of the classification problem is increased. The same trends are evident when performance is graded strictly on accuracy (S8 Fig). As cells on the periphery of each clone were not excluded from these analyses, the observed metrics provide a lower bound on the performance that may be anticipated in practice.

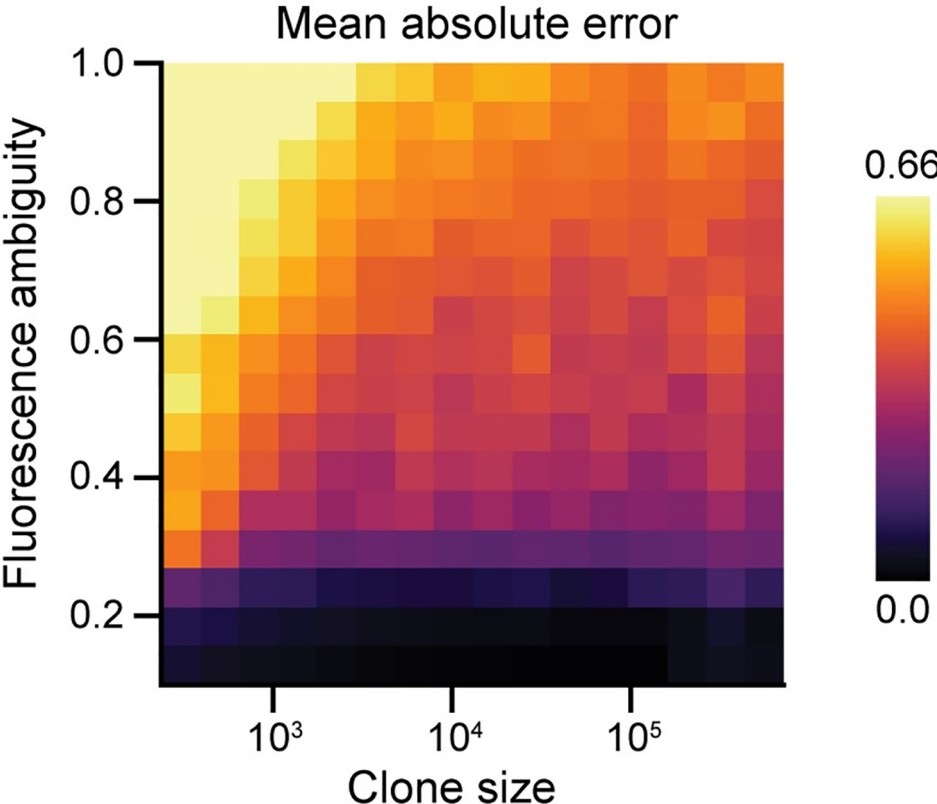

**Fig 5. Synthetic benchmarking of automated annotation performance.** Each pixel reflects the mean MAE across 50 replicates. Clone size reflects the mean number of cells per clone. Performance improves with increasing clone size and worsens with increasing fluorescence ambiguity.

Performance improved with increasing clone size. We suspected this was caused by larger clones offering additional spatial context to inform the identify of each cell. We verified our assertion by re-evaluating performance relative to a variant of our annotation algorithm that neglects spatial context (S4G Fig). As expected, the variant's performance exhibited no dependence on clone size (S9A Fig). Comparing the two strategies confirmed that spatial context confers the most benefit when clones are large (S9B Fig). Inclusion of spatial context also becomes increasingly advantageous as the fluorescence ambiguity is increased, even for smaller clones. Thus, spatial context adds progressively more value as the classification task becomes more difficult.

This observation may be rationalized from a statistical perspective. Each cell is classified by maximizing the probability that the assigned label is correct. We compute these probabilities using the estimated expression level of each cell. Neglecting spatial context, this estimate is limited to a single sample and is therefore highly sensitive to both measurement and biological noise. Incorporating spatial context expands the sample size and thereby reduces the standard error of the estimated fluorescence level. The strategy is thus generally well suited to scenarios in which fluorescence intensities correlate across large clones, and closely parallels computer vision methods that exploit spatial contiguity to segment image features with ill-defined borders [61]. Because increased measurement precision comes at the expense of spatial resolution, we expect strong performance when measurements are aggregated across relatively large clones, but failure to detect small, heterogeneous clones. These expectations are consistent

with the observed results. They are also conveniently aligned with the anticipated properties of real data, as experiments typically attempt to mitigate edge effects by driving early recombination events to generate large clones.

## Discussion

We used synthetic data to survey the performance of our annotation strategy across a much broader range of conditions than would have otherwise been possible with manually labeled data. This included conditions well beyond those of practical use. In particular, experiments designed to compare gene expression levels across clones would likely seek to avoid generating small clones with ambiguous clonal marker expression. Beyond complicating the annotation task, small clones are also exposed to diffusion-mediated signals from adjacent clones that can mask the effect of mutations. Cells located near the clone boundaries are often excluded for the same reason, as quantification is typically most reliable in cells surrounded by similar neighbors. Synthetic data provided a means to survey these edge cases and establish a lower bound on annotation performance. The strong performance observed across the remaining conditions bolsters our confidence that our annotation strategy is well suited to the images it is likely to encounter.

In each of our examples, clones were distinguished by ternary segregation of nuclear clonal marker fluorescence levels. Modern mosaic analysis techniques continue to deploy ternary labeling [62, 63], but also frequently opt for binary labeling of mutant versus non-mutant clones [64–66] and dichromic labeling of twin-spots [67, 68]. Our annotation scheme readily adapts to each of these scenarios provided that the number of anticipated labels is adjusted accordingly. In the case of dichromic labeling, binary classification would be performed separately for each color channel before merging the assigned labels. Extending the same logic to combinatorial pairs of colors suggests that our framework may also be compatible with multi-color labeling schemes used to simultaneously trace many clonal lineages over time [69–71]. A notable limitation of our approach is its reliance upon reporter fluorescence levels within distinct cells or nuclei. This requirement for discrete measurements precludes analysis of contiguous clones in which cytoplasmic fluorescence signals are indistinguishable between adjacent cells. Our framework is thus well suited to many different mosaic analysis platforms deployed in imaginal discs, so long as reporter fluorescence levels are measured on a discrete basis.

In principle, the framework described here should also be applicable to a wide variety of other tissues [72, 73] and model organisms [74–76] in which mosaics are studied. In practice, application to alternate contexts would require modifying some stages of the analysis. Most notably, image segmentation is strongly context dependent and any attempts to develop a universally successful strategy are likely to prove futile [77]. For this reason, we implemented a modular design in which each stage of analysis may be applied separately. For example, a user could perform their own segmentation before using our bleedthrough correction and clone annotation tools. By offering modular functionalities we hope to extend the utility of our software to the wider community of developmental biologists. Furthermore, the open-source nature of our framework supports continued development of more advanced features as various demands arise. Our synthetic benchmarking platform could then be used to objectively confirm the benefit conferred by any future developments.

## Materials and methods

### Genetics and microscopy of *Drosophila* eye imaginal discs

We borrowed an experimental dataset from a separate study of neuronal fate commitment during eye disc development [38]. The data consist of six eye imaginal discs dissected and

fixed during the third larval instar of *Drosophila* development. Within each disc, *ey>FLP* and *FRT40A* were used to generate clones. The chromosome arm (2L) targeted for recombination was marked with a *Ubi-mRFPnls* transgene (S1A Fig), enabling automated detection of clones marked by distinct levels of mRFP fluorescence (S1B Fig). The discs also carried a *pnt-GFP* reporter transgene located on a different chromosome that was not subject to mitotic recombination. The PntGFP reporter is predominantly expressed in two narrow stripes of progenitor cells during eye disc development [38]. The first stripe occurs immediately posterior to a wave of developmental signaling that traverses the eye disc. Progenitor cells located in this region are suitable for comparison because they are of approximately equivalent developmental age. We applied the Fly-QMA framework to a total of nine images of these cells.

Genetics, fly lines, immunohistochemistry, and imaging conditions related to this dataset have already been published [38]. All discs were dissected in PBS, fixed in 4% paraformaldehyde for 30 min at room temperature, and permeabilized with PBS-Triton X-100 0.1% for 20 min at room temperature to allow DAPI penetration without perturbing the fluorescence of the Pnt-GFP protein. Discs were subsequently stained with a 4',6-diamidino-2-phenylindole (DAPI) nuclear marker, rinsed twice with PBS-Tween 0.5%, and mounted on Vecta Shield (Vector labs). Images were acquired using a Leica SP5 confocal equipped with a tunable detector. The 405, 488, and 561 nm lasers were used to excite DAPI, Pnt-GFP, and Ubi-mRFPnls, while photons were collected in the 437-481, 491–555, and 570-644 nm intervals for DAPI, GFP, and mRFP, respectively. Images were recorded with 16-bit resolution using a 40X oil objective. Discs were oriented with the dorso-ventral equator parallel to the horizontal axis, and all images captured at least six rows of ommatidia on either side of the equator. All discs were fixed, mounted, and imaged in parallel in order to reduce measurement error.

## Characterization of fluorescence bleedthrough

For each image, we morphologically dilate the foreground until no features remain visible (S2A Fig). We then extract the background pixels and resample them such that the distribution of pixel intensities is approximately uniform (S2B Fig). Resampling helps mitigate the skewed distribution of pixel intensities found in the background. We then estimate values for each $\{\alpha_1, \alpha_2, \ldots \alpha_K\}$ and $\beta$ by fitting a generalized linear model to the fluorescence intensities of the resampled pixels (S2C Fig). Each model is a variant of Eq 3 in which angled braces instead denote averages across all background pixels. We formulate these models with identity link functions under the assumption that residuals are gamma distributed. Their coefficients provide an estimate of the bleedthrough contribution strengths that may then be used to estimate the background fluorescence intensity of each nucleus in the corresponding image (S2D Fig). The measurements may then be corrected through application of Eq 4.

## Clone annotation algorithm

We assume the measured fluorescence level $x_i$ for cell $i$ is sampled from an underlying distribution $p_m(x)$ for cells carrying $m$ copies of the gene encoding the clonal marker:

$$x_i \sim p_m(x) \tag{5}$$

We further assume that $p_m(x)$ is comprised of a mixture of one or more lognormal distributions:

$$p_m(ln\ x) = \sum_{n=1}^{N} \lambda_n \mathcal{N}(ln\ x | \theta_n) \tag{6}$$

$$\sum_{n=1}^{N}\lambda_n = 1 \tag{7}$$

where $0 \leq \lambda \leq 1$ are the mixing proportions, $\theta_n = (\mu_n, \sigma_n^2)$ are the mean and variance of the $n$th distribution. This assumption is supported by both empirical observations and theoretical insights [46, 47]. By superposition, the global distribution of measured fluorescence levels $p(lnx)$ for all values of $m$ are also sampled from a mixture of $K$ components:

$$p(ln\ x) = \sum_{m=0}^{2}\alpha_m p_m(ln\ x) = \sum_{m=0}^{2}\alpha_m\sum_{n=1}^{N}\lambda_n\mathcal{N}(ln\ x|\theta_n) = \sum_{k=1}^{K}\lambda_k\mathcal{N}(ln\ x|\theta_k) \tag{8}$$

$$\sum_{k=1}^{K}\lambda_k = 1 \tag{9}$$

where $\alpha_m$ denotes the overall fraction of cells with $m$ copies of the gene encoding the clonal marker. For brevity, we substitute $X = lnx$ yielding:

$$p(X) = \sum_{k=1}^{K}\lambda_k\mathcal{N}(X|\theta_k) \tag{10}$$

Given a collection of sampled fluorescence levels, $\{X_i\}_{i\ =\ 1...N}$, we use expectation maximization to find values of $\theta_k$ and $\lambda_k$ for each of the model's $K$ components that maximize the log-likelihood of the observed sample. We repeat this procedure for a range of sequential values of $K$, resulting in multiple models of increasing size. We then balance model resolution against overfitting by selecting the model that yields the smallest value of the Bayesian Information Criterion (BIC):

$$BIC(K) = ln(N)q_K - 2ln(\hat{L}_K) \tag{11}$$

$$q_K = K - 1 + 2^K \tag{12}$$

where $N$ is the sample size, $ln(\hat{L})_K$ is the maximum value of the log-likelihood, the subscript $K$ denotes the number of mixture components in the model, and $q_K$ is the total number of parameters (i.e. $K - 1$ values of $\lambda_k$ and $2^K$ values of $\mu_k$ and $\sigma_k^2$).

Applying Bayes' rule to the selected model infers the posterior probabilities that each sample $X_i$ belongs to the $k$th component:

$$p(k|X_i) = \frac{p(X_i|k)p(k)}{p(X_i)} = \frac{p(X_i|k)\lambda_k}{p(X_i)} \tag{13}$$

where $p(X_i|k)$ is evaluated using the model's likelihood function and $p(X_i)$ is evaluated by marginalizing across each of the model's $K$ components. The end result is a mixture model that allows us to predict the probability that a given measurement of clonal marker expression belongs to a particular one of its component distributions.

We then define a many-to-one mapping, $f$, from each of the $K$ components of the mixture to each of the three possible values of $m$:

$$f : \{0, 1, \ldots K\} \rightarrow \{0, 1, 2\} \tag{14}$$

We determine the mapping by k-means clustering the $K$ component distributions into three groups on the basis of their mean values, $e^{\mu_k}$. We may then assign a genotype label $m$ to each measurement $X_i$ by predicting the component $k$ from which it was sampled.

The accuracy of these labels depends upon how closely the fitted mixture model reflects the true partitioning of gene copies among clones. While finite mixtures are always identifiable given a sufficiently large sample [48], the algorithm used to fit the mixture tends toward local maxima of the likelihood function when the true components are similar (Wu, 1983). An approach based on a univariate mixture is thus inherently prone to failure when expression levels extensively overlap across clones, as variation within each clone precludes accurate classification of a cell's genotype solely on the basis of its individual expression level. However, clonal lineages are unlikely to exist in isolation because recombination events are usually timed to generate large clones. Our strategy therefore integrates both clonal marker expression and spatial context to identify clusters of cells with locally homogeneous expression behavior.

We incorporate spatial context by introducing a second jointly-distributed variable $Y_i$:

$$Y_i = \frac{1}{M_i} \sum_{j=0}^{M_i} X_j \tag{15}$$

where the subscript $j$ indexes all $M_i$ neighbors of cell $i$. The new variable reflects the average expression level among the neighbors surrounding each cell. We define neighbors as pairs of cells located within a critical distance of each other. This distance, or sampling radius, is derived from the approximate length scale over which cells retain approximately similar clonal marker expression levels. Specifically, we determine the exponential decay constant of the spatial correlation function, $\psi(\delta)$:

$$\psi(\delta) = \frac{< (X_i - \mu_X)(X_j - \mu_X)>_{i,j \in \delta}}{\sigma_X^2} \tag{16}$$

where $\mu_X$ and $\sigma_X^2$ are the global mean and standard deviation, and angled brackets denote the mean across all pairs of cells separated by distance $\delta$. We efficiently implement this procedure by fitting an exponential decay function to the down-sampled moving average of $\psi(\delta)$ as a function of increasing separation distance.

Following the introduction of spatial context, the mixture model becomes:

$$p(X, Y) = \sum_{k=1}^{K} \lambda_k \mathcal{N}(X, Y | \theta_k) \tag{17}$$

where $\theta_k = (\vec{\mu}_k, \vec{\sigma}_k^2)$ contains the mean and variance of each component given by vectors of length two. This formulation constrains each component's covariance matrix to be diagonal. The posterior is now:

$$p(k | X_i, Y_i) = \frac{p(X_i, Y_i | k) \lambda_k}{p(X_i, Y_i)} \tag{18}$$

We can recover the univariate model by marginalizing the posterior over all values of $Y$:

$$p(k | X_i) = \sum_j p(k | X_i, Y_j) \tag{19}$$

When neglecting spatial context, we use this expression to classify each sample by applying the mapping $f$ to the value of $k$ that maximizes $p(k|X_i)$:

$$f(\underset{k}{\mathrm{argmax}}\ p(k|X_i)) \tag{20}$$

In all other cases, we deploy a graph-based approach to refine the estimate of $p(k|X_i, Y_i)$. This first entails constructing an undirected graph connecting adjacent cells within each image. We obtain the graph's edges through Delaunay triangulation of the measured cell positions, then exclude distant neighbors by thresholding the edge lengths. Each edge is assigned a weight $w_{ij}$ reflecting the similarity of clonal marker expression between adjacent cells $i$ and $j$:

$$w_{ij} = exp(\frac{-E_{ij}}{\langle E \rangle}) \tag{21}$$

$$E_{ij} = |X_i - X_j| \tag{22}$$

where $E_{ij}$ is the absolute log fold-change in measured expression level and angled brackets denote the mean across all edges. We chose an exponential formulation because it yields an approximately uniform distribution of edge weights. We then detect communities within the graph using the Infomap algorithm [49]. The algorithm provides a hierarchical partitioning of nodes into non-overlapping clusters. We aggregate all clusters below a critical level that is again chosen by estimating the spatial correlation decay constant. We then enumerate $p(k \mid X_i, Y_i^c)$ where $Y_i^c$ is the spatial context obtained by averaging expression levels among all neighbors in the same community as cell $i$.

We further incorporate spatial context by allowing the posterior probabilities $p(k \mid X_i, Y_i^c)$ to diffuse among adjacent cells. We define the modified posterior probability $\hat{p}(k \mid X_i, Y_i^c)$ through a recursive relation analogous to the Katz centrality [50], initialized by $p(k \mid X_i, Y_i^c)$:

$$\hat{p}(k \mid X_i, Y_i^c) = \alpha \sum_j w_{ij}\hat{p}(k \mid X_i, Y_i^c) + \beta \tag{23}$$

$$\beta = (1 - \alpha)p(k|X_i, Y_i^c) \tag{24}$$

where $\alpha$ is the attenuation factor and $w_{ij}$ are the edge weights. Expressed in matrix form, the solution for $\hat{p}(k \mid X, Y^c)$ is given by:

$$\hat{p}(k \mid X, Y^c) = (I - \alpha W)^{-1}(1 - \alpha)p(k \mid X, Y^c) \tag{25}$$

where $I$ denotes the identity matrix and $W$ is the matrix of edge weights $w_{ij}$. We then assign a label to each measurement $X_i$ by applying $f$ to the value of $k$ that maximizes $\hat{p}(k \mid X_i, Y_i^c)$:

$$f(\underset{k}{\mathrm{argmax}}\ \hat{p}(k|X_i, Y_i^c)) \tag{26}$$

Finally, we assess the total posterior probability of each assigned label, $\hat{P}(m_i)$:

$$\hat{P}(m_i) = \sum_{\{k|f(k)=m_i\}} \hat{p}(k|X_i, Y_i^c) \tag{27}$$

This measure reflects the overall confidence that $m_i$ is the appropriate label. Labels whose confidence falls below 80% are replaced by their counterparts estimated using the marginal classifier. This substitution helps preserve classification accuracy in situations where spatial

context is not informative, and is particularly useful when the annotated clones are relatively small.

## Statistical comparison of fluorescence levels

To mitigate edge effects, cells residing on the periphery of each clone were excluded from all comparisons (S2E Fig). Border cells were identified by using a Delaunay triangulation to find all cells connected to a neighbor within a different clone. Our framework includes a simple graphical user interface that permits manual curation of which regions of the image field are included in subsequent analyses. We used this tool to limit our analysis to the region of elevated GFP expression near the morphogenetic furrow (S2F Fig). Comparisons were further restricted to cells undergoing similar stages of development (S2G Fig). These restrictions served to buffer against differences in developmental context and ensured that all compared cells were of similar developmental age. The remaining fluorescence measurements were then aggregated across all eye discs and compared between pairs of clones by two-sided Mann-Whitney *U* test.

## Simulated cell growth and recombination

We simulated the two dimensional growth of a cell culture seeded with a single cell. Growth proceeds through sequential division of cells (S6A Fig). Not all cells divide at each time-step because cell division is a stochastic process. Instead, each cell divides stochastically with a rate controlled by a global growth rate parameter.

Cells in this culture carry a gene encoding a clonal marker (S6B Fig). During growth, the gene is subject to mitotic recombination (S6C Fig). Each time a cell divides, its genes are duplicated and equally partitioned between the two daughter cells. However, in some instances a heterozygous parent may instead partition its two duplicate genes unequally, with one daughter receiving both and the other receiving none. These mitotic recombination events occur stochastically with a frequency defined by a global recombination rate parameter.

After each round of cell division, all cells are repositioned in order to preserve approximately uniform spatial density (S6C Fig). Repositioning is achieved by equilibrating a network of springs connecting each cell with its neighbors. This undirected network is constructed through Delaunay triangulation of all cells spatial positions. Edges on the periphery of the culture are systematically excluded by establishing a maximum polar angle between neighbors. This filtration removes spurious edges between distant pairs of cells. Edges connecting pairs of cells with the same clonal marker dosage are assigned a 10% higher spring constant than edges that connect dissimilar cells. This modest bias ensures that cells tend to remain proximal to their clonal lineages. Cell positions are then updated using a force-directed graph drawing algorithm [51]. Alternating cell division and repositioning steps are then repeated until a predefined population size is reached.

The timing and duration of recombination events affects the number and size of the resultant clones. In real experiments, recombination events are restricted to a particular stage of the developmental program through localized exogenous expression of the recombination machinery. We incorporated this feature into our cell growth simulations via two adjustable parameters. The first determines the minimum population size at which recombination may begin, while the second determines the number of generations over which recombination may continue to occur. These two parameters provide a means to tune the average number and size of clonal subpopulations in the synthetic data (S6D Fig). Early recombination events generally entail larger clones, while shorter recombination periods limit the extent of clone formation (S6E Fig).

## Generation of synthetic microscopy data

Each simulation yields a list of spatial coordinates and gene dosages for each nucleus (S6B Fig). Synthetic measurements for each nucleus were generated by randomly sampling fluorescence levels $\{x_1, x_2, \ldots x_{i=N}\}$ from a lognormal distribution conditioned upon the corresponding gene dosage (S7A–S7C Fig):

$$ln\ x \sim \mathcal{N}_n(\theta_n) \tag{28}$$

where the subscript $n$ denotes the gene copy number and $\theta_n = (\mu_n, \sigma_\alpha^2)$ are the mean and variance of the corresponding distribution. We define $\mu_n$ such that the mean fluorescence level doubles for each additional copy of the gene:

$$\mu_n = ln(2^{n-1}) \tag{29}$$

We refer to $\sigma_\alpha$ as the *fluorescence ambiguity* because it modulates the similarity of fluorescence levels across gene dosages. Increasing $\sigma_\alpha$ increases the overlap among $\mathcal{N}_0$, $\mathcal{N}_1$, and $\mathcal{N}_2$ (S7D and S7E Fig), and consequently increases the difficulty of the annotation task (S7F Fig).

## Synthetic benchmarking of annotation performance

We generated a large synthetic dataset spanning a broad range of sixteen different clone sizes and fluorescence ambiguities (S6D and S7F Figs, only half are shown). We performed 50 replicate simulations for each condition. All simulations were terminated when the total population exceeded 2048 cells. We assigned each cell a 20% probability of division upon each iteration, and each cell division event was accompanied by a 20% chance of mitotic recombination. Parent cells containing zero or two copies of the recombined genes were ineligible for recombination, effectively sealing the genetic fates of their respective lineages.

To annotate each set of measurements, the mixture model given by Eq 17 was independently trained and applied to each replicate. Training a single model on all replicates yields modestly stronger performance on average, but also yields more variable variable results across the parameter space because all labels are dependent upon the outcome of a single expectation maximization routine.

## Data and software availability

We have distributed the automated mosaic analysis framework as an open-source python package available at https://sebastianbernasek.github.io/flyqma. The associated code repository contains resources designed to help users analyze their own microscope images. These include code documentation, a guide to getting started with Fly-QMA, and an interactive tutorial that uses example data to demonstrate the core features of the software. We also intend to incorporate Fly-QMA into future versions of *FlyEye Silhouette*, our open-source desktop application for quantitative analysis of the larval eye. The code used to generate synthetic microscopy data is also freely available at https://github.com/sebastianbernasek/growth. All segmented and annotated eye discs are accessible via our data repository (https://doi.org/10.21985/N2F207).

## Supporting information

**S1 Fig. Example clones in the larval fly eye.** (A) Genetic schema for a bleedthrough control experiment. Red and green ovals represent genes encoding a RFP-tagged clonal marker and a GFP-tagged control reporter, respectively. Black lines depict a genomic locus. Recombination does not affect gene dosage of the control reporter, so GFP variation across clones is attributed

to fluorescence bleedthrough. (B) Confocal image of an eye imaginal disc. Red, green, and blue reflect clonal marker, control reporter, and nuclear stain fluorescence, respectively. (C) Segmentation of the DAPI nuclear stain. White lines show individual segments.
(TIF)

**S2 Fig. Using background pixels to characterize bleedthrough contributions in the foreground.** (A) Extraction of background pixels (striped region). Foreground includes the merged RFP and GFP images, surrounded by a white line. White arrow marks the morphogenetic furrow (MF). (B) Background pixel values are resampled such that RFP intensities are uniformly distributed. (C) A generalized linear model characterizes the contribution of RFP bleedthrough to GFP fluorescence. Boxes reflect windowed distributions of resampled background pixel intensities. Red line shows the model fit. (D) Measured GFP levels before bleedthrough correction. Markers represent individual nuclei. Red line shows the inferred contributions of RFP fluorescence bleedthrough. Dashed portion is extrapolated. (E-G) Data curation prior to statistical comparison of GFP levels. (E) Cells on the periphery of each clone are excluded. (F) The selection is limited to the region of elevated GFP expression near the MF. (G) It is further limited to cells of the same developmental age, defined by their relative positions along the x-axis.
(TIF)

**S3 Fig. Training a clone annotation model.** (A) One or more images are segmented, yielding a set of fluorescence measurements $X$. These are used to sample the spatial context $Y$ of the neighborhood surrounding each cell. Both sets of values are used to train a mixture model. Subsequent panels demonstrate these procedures using the example shown in S3 Fig C. (B) Expression levels are jointly distributed with the local average among neighboring cells. Center panel shows the joint distribution. Top and right bar plots show marginal distributions. (C) Mixture model identifies seven distinct components $k_i$. Center panel shows position and spread of each component. Top and right panels show marginal components scaled by their respective weights. Red shading denotes the label $m_i$ assigned to each component. The model predicts the posterior probabilities that a given sample $(X, Y)$ belongs to each component. (D) Neighborhood size is estimated by computing the decay constant of the spatial correlation function, $\psi(\delta)$. Black line shows the moving average of $\psi(\delta)$, red line shows an exponential fit. Inset shows the resultant sampling region. (E) The optimal number of mixture components is determined by minimizing BIC score. (F) Mixture components are labeled by k-means clustering their mean values. Markers reflect the component means, colors denote the assigned label.
(TIF)

**S4 Fig. Label assignment using a trained clone annotation model.** (A) Measurements are used to sample spatial contexts before the trained model is applied (blue and green path). In parallel, measurements are labeled using a marginal projection of the trained model (magenta path). The labels are then merged (red path). (B-D) Spatial context sampling. (B) Weighted undirected graph connecting adjacent cells. Line width reflects expression similarity between neighbors. (C) Community resolution is defined by aggregating clusters that fall below a hierarchical cut level $\delta$. Panels show increasing levels of aggregation. Colors denote distinct communities. (D) Cut level is chosen by finding the maximum level (red dot) that remains lower than the decay constant of the spatial correlation function, $\psi(\delta)$ (black line). Panel E depicts aggregation below the third level for ease of visualization. (E) Application of the mixture model. *(I)* The graph contains distinct communities of locally similar expression. *(II)* Mean expression level within each community serves as the local average for each cell. *(III)* Mixture model estimates the probability that each cell belongs to each of its component. Bar plots

within each cell illustrate the cumulative probability of each label. *(IV)* Posterior probabilities are diffused across the graph. *(V)* Each cell is assigned the most probable label. (F,G) Application of a marginal mixture model. (F) Marginal mixture components, shaded by their mapped labels. Dashed line is the overall marginal density. (G) Marginal classifier labels cells strictly on the basis of their individual fluorescence level. Red shading denotes the most probable label for each level. (H) Annotated measurements. Red shading denotes the assigned label. Labels with low confidence $\hat{P}(m_i) < 0.8$ are replaced by their marginal counterparts.
(TIF)

**S5 Fig. Comparison of automated annotation with manually assigned labels.** (A) Distribution of labels among each possible value. (B) Visual comparison of the sole instance in which automated and manual annotation differ. Image shows clonal marker fluorescence, colors denote the assigned label.
(TIF)

**S6 Fig. Simulated growth of a synthetic cell culture.** (A) Partial simulation time course. Each marker depicts a cell. Greyscale intensity reflects clonal marker gene dosage. Simulation time reflects the approximate number of cell divisions since the initial seed. (B) Simulations yield gene dosages and spatial coordinates for each cell. (C) Single iteration of an example simulation. Circles represent individual cells, red shading denotes clonal marker dosage. Cycles of cell division, recombination, and repositioning are repeated until the simulation reaches a specified end time ($t > 11$ in panel A). (D) Cultures simulated with varying recombination start times. All cultures were subject to four generations of recombination ($\delta t = 4$). Recombination start time increases from left to right. Later recombination events generally yield smaller clones. (E) Mean clone size (cells per clone) as a function of the recombination start time. Colors denote recombination period duration. Error bars reflect standard error of the mean across 50 replicates. Clone size generally decreases as recombination is limited to later times.
(TIF)

**S7 Fig. Tunable generation of synthetic microscopy data.** (A) Fluorescence levels are sampled from lognormal distributions conditioned upon gene dosage. (B) Synthetic data include a measured fluorescence level for each reporter in each cell. Text color reflects the generative distribution in A. (C) Synthetic image of clonal marker fluorescence when $\sigma_\alpha = 0.25$. Each nucleus is shaded in accordance with its sampled fluorescence intensity. (D-F) Left to right, increasing the fluorescence ambiguity parameter broadens the overlap in fluorescence levels across gene dosages. (D) Distributions used to generate clonal marker fluorescence levels. Red shading denotes gene dosage. (E) Evenly weighted sum of the generative distributions. (F) Example images of clonal marker fluorescence.
(TIF)

**S8 Fig. Fraction of nuclei correctly labeled during synthetic benchmarking.** Each pixel reflects the average across 50 replicates. Clone size reflects the mean number of cells per clone. Performance improves with increasing clone size and worsens with increasing fluorescence ambiguity.
(TIF)

**S9 Fig. Spatial context is most informative for large clones with ambiguous fluorescence.** (A) MAE of labels assigned using a marginal classifier that neglects spatial context. Performance worsens with increasing fluorescence ambiguity but does not depend upon clone size. (B) Annotation performance relative to the marginal classifier. Color scale reflects the $\log_2$

fold-change in MAE when spatial context is neglected. Blue indicates that spatial context improves performance.
(TIF)

## Author Contributions

**Conceptualization:** Sebastian M. Bernasek, Luís A. N. Amaral.

**Data curation:** Sebastian M. Bernasek, Nicolás Peláez.

**Formal analysis:** Sebastian M. Bernasek.

**Funding acquisition:** Richard W. Carthew, Neda Bagheri, Luís A. N. Amaral.

**Investigation:** Nicolás Peláez.

**Methodology:** Sebastian M. Bernasek, Nicolás Peláez, Luís A. N. Amaral.

**Project administration:** Richard W. Carthew, Neda Bagheri, Luís A. N. Amaral.

**Resources:** Richard W. Carthew, Neda Bagheri, Luís A. N. Amaral.

**Software:** Sebastian M. Bernasek.

**Supervision:** Richard W. Carthew, Neda Bagheri, Luís A. N. Amaral.

**Validation:** Sebastian M. Bernasek.

**Visualization:** Sebastian M. Bernasek.

**Writing – original draft:** Sebastian M. Bernasek.

**Writing – review & editing:** Sebastian M. Bernasek, Nicolás Peláez, Richard W. Carthew, Neda Bagheri, Luís A. N. Amaral.

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
