## [Decision Letter · Decision Letter 0]

13 Dec 2019

Dear Dr Bernasek,

Thank you very much for submitting your manuscript, 'Fly-QMA: Automated analysis of mosaic imaginal discs in Drosophila', to PLOS Computational Biology. As with all papers submitted to the journal, yours was fully evaluated by the PLOS Computational Biology editorial team, and in this case, by independent peer reviewers. The reviewers appreciated the attention to an important topic but identified some aspects of the manuscript that should be improved.

We would therefore like to ask you to modify the manuscript according to the review recommendations before we can consider your manuscript for acceptance. Your revisions should address the specific points made by each reviewer and we encourage you to respond to particular issues Please note while forming your response, if your article is accepted, you may have the opportunity to make the peer review history publicly available. The record will include editor decision letters (with reviews) and your responses to reviewer comments. If eligible, we will contact you to opt in or out.raised.

- Supporting Information uploaded as separate files, titled 'Dataset', 'Figure', 'Table', 'Text', 'Protocol', 'Audio', or 'Video'.

We hope to receive your revised manuscript within the next 30 days. If you anticipate any delay in its return, we ask that you let us know the expected resubmission date by email at ploscompbiol@plos.org.

Sincerely,

Pedro Mendes, PhD

Associate Editor

PLOS Computational Biology

Douglas Lauffenburger

Deputy Editor

PLOS Computational Biology

[LINK]

Reviewer's Responses to Questions

**Comments to the Authors:**

Reviewer #1: The authors have developed “Fly-QMA”, an unsupervised annotation computational algorithm for automating clonal analysis on confocal microscopy images of Drosophila imaginal discs. This could be a very useful program for many researchers in Drosophila, and apparently also could be used to analyze clones in other models as well. The authors tested real and synthetic data to validate the efficacy of the annotation algorithm across different conditions. Although I lack expertise in the mathematics of the work, my take on it is that it could be a useful tool.

Strengths of the work include:

• Uses nuclear fluorescence quantification in segmentation

• Automated bleed-through correction

• Modular arrangement allows flexibility eg, different tissues or contexts

• Unsupervised annotation of clones, using clonal marker expression info and spatial information

• Should work well if markers are clear and different (eg membrane versus nuclear?)

• Best performance with larger clones (as is true in any method of analyzing clones).

• The algorithm is available on github, with a plan to incorporate core aspects into FlyEye Silhouette (their open source platform).

There are a few weaknesses, as well, although relatively minor:

• operating this algorithm seems complicated for someone who is not savvy with computer programming –even installing requires Python 3.6+. Even the tutorial seems complicated. Although some will be able to use it as is, it would be even more useful if the algorithm could be incorporated into a user-friendly and widely used program like Fiji/Image J.

• Requires use of a 3D confocal stack, which is fine for some analyses but can it also be used for single images from epifluorescence scope (e.g., +GFP vs -GFP)?

• Figure S8, “Fraction of nuclei correctly labeled during synthetic benchmarking” appears to be missing from the manuscript (the image labelled S8 appears to be S9, according to the legend information).

Reviewer #2: Bernasek et al. present a new method for automatically quantifying fluorescence reporter expression of cells in Drosophila imaginal discs using confocal microscopy. The proposed pipeline starts with the automatic segmentation of the image data based on well-known methods like CLAHE for local contrast enhancement, Otsu’s method for image binarization and a combination of Euclidean distance maps and a seeded watershed to identify distinct connected components for each of the cell nuclei. Moreover, a bleedthrough correction is performed that tries to eradicate the crosstalk impact of other fluorescent reporters or due to autofluorescence. Cells are then classified into three distinct expression levels, including a neighborhood analysis to correct for outliers. The analysis software and the simulated benchmark are released as open source, which definitely will increase the potential usefulness to the community. Moreover, the software contains an extensive documentation, getting started sections and tutorials that should make it straightforward to use the tool (given that at least basic knowledge of Python usage is available). The paper is very well written and I only have a few comments.

Comments:

- The author summary partly contains exact replications of the abstract. This should be avoided and at least a slight rephrasing should be performed.

- The notation with <>-brackets for the average across all pixels seems unusual to me. Maybe rather use a horizontal bar on top of the variable letter?

- To better understand that manual assessment is infeasible if the data set sizes grow, it would be good to have a comparison of the „processing times“ of a human expert and the software.

- If I understood it correctly, the simulated benchmark does not involve varying contrast in the different image regions. An easy addition for increased realism would be to add global illumination artifacts to the simulated images in order to also assess the validity of the CLAHE approach and to see if it works as expected for the simulated scenario.

- The reference list mixes sentence case and title case for the titles and journal/conference names. While being a minor issue, I would recommend to use either one or the other but not mixing them.

- Page 10, Fig. 2C should probably be Fig. 3C instead?

Reviewer #3: I will comment on the biology and image analysis. I am not qualified to review the mathematical equations critically and therefore hope that these will be covered by another reviewer.

The authors describe Fly-QMA a computational tool for performing quantitative clonal analysis on mosaicly labelled drosophila imaginal discs. The authors test their software using both real and synthetic data and demonstrate a robust performance. These tools are important and will be increasingly in demand and so I commend any effort to move the field forward. I have a number of questions that I don’t feel come out in the main text that could therefore be clarified.

1). Whilst bleed through correction might be a useful feature it would be pertinent to eliminate bleed through at the image acquisition stage by careful choice of excitation and emission filters or tuning of detectors. It is not possible to judge from the data presented what the cause of the bleed through is because details of the image acquisition are not included in the methods. The authors should add this detail. I would expect to see it here in this case even though it is previously published.

2). Historically the main bottleneck in automated clonal analyses (especially of data from fluorescent reporters) is in segmentation. This is often best achieved by trial and error because of the large variability between datasets and in their acquisition parameters. The authors are wise to point out that the pipeline can accommodate images segmented before hand but give no guidance on what the requirement of their software might be? Do they require the segmentation masks or regions of interest? Where is the standardised file structure described is it on github? Please clarify?

3). Non of the microscopy images in any of the figures include scale bars and it is not clear what objective was used for image capture so it is hard to tell what a comparable system for image acquisition would be. Please revise all the figures to include scale bars.

4). The analysis appears to be limited to nuclear signals as this segmentation occurs on nuclei. This is a limitation as often the analysis is designed to identify contiguous clones for which the extent of the cytoplasm may be important. This should be discussed.

5). I don’t see is any attempt to derive the size (spatially or by number of cells) of the individual clones? Or to correct for the probability of multiple clones being adjacent to one another?

Size is a key parameter and this is a common problem in many clonal analyses. Does the variation in intensities between clones with the same copy number mean that the algorithm can discriminate between adjacent clones? This seems unlikely. Therefore can the clone sizes be corrected for the probability of being multiple clones?

Or, with the spatial information available, can large polyclonal units be identified in the annotation?

6) The authors don’t do the excellent resources provided on github justice - please make it clear that there are tutorials and test data available there.

**Have all data underlying the figures and results presented in the manuscript been provided?**

Reviewer #1: No: Figure S8 appears to be missing

Reviewer #2: Yes

Reviewer #3: Yes

PLOS authors have the option to publish the peer review history of their article (what does this mean?). If published, this will include your full peer review and any attached files.

Reviewer #1: No

Reviewer #2: No

Reviewer #3: No

---

## [Decision Letter · Decision Letter 1]

27 Jan 2020

Dear Dr Bernasek,

We are pleased to inform you that your manuscript 'Fly-QMA: Automated analysis of mosaic imaginal discs in Drosophila' has been provisionally accepted for publication in PLOS Computational Biology.

Before your manuscript can be formally accepted you will need to complete some formatting changes, which you will receive in a follow up email. A member of our team will be in touch within two working days with a set of requests.

Best regards,

Pedro Mendes, PhD

Associate Editor

PLOS Computational Biology

Douglas Lauffenburger

Deputy Editor

PLOS Computational Biology

Reviewer's Responses to Questions

**Comments to the Authors:**

Reviewer #3: The authors have satisfactorily addressed the comments and concerns that I raised.

**Have all data underlying the figures and results presented in the manuscript been provided?**

Reviewer #3: Yes

PLOS authors have the option to publish the peer review history of their article (what does this mean?). If published, this will include your full peer review and any attached files.

Reviewer #3: No

---

## [Editor Report · Acceptance letter]

25 Feb 2020

PCOMPBIOL-D-19-01579R1 

Fly-QMA: Automated analysis of mosaic imaginal discs in Drosophila

Dear Dr Bernasek,

I am pleased to inform you that your manuscript has been formally accepted for publication in PLOS Computational Biology. Your manuscript is now with our production department and you will be notified of the publication date in due course.

With kind regards,

Laura Mallard
